# Proteomics- and Metabolomics-Based Analysis of Metabolic Changes in a Swine Model of Pulmonary Hypertension

**DOI:** 10.3390/ijms24054870

**Published:** 2023-03-02

**Authors:** Payel Sen, Bachuki Shashikadze, Florian Flenkenthaler, Esther Van de Kamp, Siyu Tian, Chen Meng, Michael Gigl, Thomas Fröhlich, Daphne Merkus

**Affiliations:** 1Walter Brendel Center for Experimental Medicine (WBex), University Clinic Munich, LMU Munich, 81377 Munich, Germany; 2Center for Cardiovascular Research (DZHK), Munich Heart Alliance (MHA), Partner Site Munich, 81377 Munich, Germany; 3Laboratory for Functional Genome Analysis (LAFUGA), Gene Center, LMU Munich, 81377 Munich, Germany; 4Division of Experimental Cardiology, Erasmus University Medical Center, 3015 GD Rotterdam, The Netherlands; 5Bavarian Center for Biomolecular Mass Spectrometry, Technical University Munich, 85354 Freising, Germany

**Keywords:** pulmonary hypertension, proteomic analysis, metabolomic analysis

## Abstract

Pulmonary vein stenosis (PVS) causes a rare type of pulmonary hypertension (PH) by impacting the flow and pressure within the pulmonary vasculature, resulting in endothelial dysfunction and metabolic changes. A prudent line of treatment in this type of PH would be targeted therapy to relieve the pressure and reverse the flow-related changes. We used a swine model in order to mimic PH after PVS using pulmonary vein banding (PVB) of the lower lobes for 12 weeks to mimic the hemodynamic profile associated with PH and investigated the molecular alterations that provide an impetus for the development of PH. Our current study aimed to employ unbiased proteomic and metabolomic analyses on both the upper and lower lobes of the swine lung to identify regions with metabolic alterations. We detected changes in the upper lobes for the PVB animals mainly pertaining to fatty acid metabolism, reactive oxygen species (ROS) signaling and extracellular matrix (ECM) remodeling and small, albeit, significant changes in the lower lobes for purine metabolism.

## 1. Introduction

Pulmonary hypertension (PH) due to pulmonary vein stenosis (PVS) is a life-threatening disease, which mainly affects the pediatric population [1]. This type of PH, which ultimately results in a left ventricular inflow tract obstruction, is classified under type II PH [2]. PVS presents mostly with congenital heart defects (univentricular heart disease, ventricular septal defect, atrial septal defect or persistent arterial duct), lung disease (bronchopulmonary dysplasia) or Down syndrome or other trisomy [3,4]. In rare cases, PVS can also occur in adults because of radiofrequency ablation therapy after atrial fibrillation [5]. This particular type of PH is characterized by an initial passive increase in pulmonary artery pressure brought on by increased resistance due to the banding. The increased mean pulmonary artery pressure results in vascular remodeling, which further raises pulmonary vascular resistance and causes an additional increase in pressure. Surgical interventions and/or stenting of the lesions in patients with PVS frequently lead to restenosis, and the use of vasodilators comes with the risk for pulmonary edema [6]. The complex molecular mechanisms involved in PH are limiting factors in the development of novel therapeutic interventions.

Previous work by our group has shown that endothelial factors are important in the development of PH in a swine model using pulmonary vein banding (PVB) of the lower lobes for 12 weeks to induce type II PH [7]. This procedure results in areas of the lung with a varied hemodynamic profile within the lung; the lower lobes experience high pressure and low flow (HF/LF) whereas the upper lobes experience high pressure and high flow (HP/HF). High and low shear stress have very striking effects on the endothelial cells of the lung vasculature. Endothelial cells typically respond to high shear stress with strong nitric oxide synthesis, but they “activate” a pro-inflammatory profile at low shear stress, characterized by low nitric oxide production [8]. In this study, we conducted a quantitative LC-MS/MS-based proteomic analysis of lung samples along with untargeted metabolomics from swine with PVB and control (Cntrl) swine. We analyzed tissues from the upper and lower lobes to investigate how different hemodynamic profiles impact protein and metabolite expression due to PH in the lobes.

## 2. Results

### 2.1. Characteristics of Pulmonary Hypertension

Pulmonary vein banding in the PVB group animals resulted in significant stenosis in the inferior pulmonary confluence as shown in the angiogram (Figure 1A). Twelve weeks after banding, this resulted in a significantly higher mean pulmonary artery pressure (38 ± 8 mmHg) in the PVB animals compared to the control (mean of 20 ± 4 mmHg, *p* < 0.05) (Figure 1D) as well as an increased pulmonary vascular resistance and reduced pulmonary artery compliance (Figure 1E,F).

Histology of the lung tissue revealed more picrosirius red staining in the PVB animals in the upper and lower lobe, depicting more fibrosis compared to the Cntrl (Figure 1B,C).

### 2.2. Proteomic Analysis of PVB vs. Control in the Upper and Lower Lobe

To explore the chronic effects of flow and pressure alterations in the lung tissue, we performed a label-free liquid chromatography–tandem mass spectrometry analysis (LC-MS/MS) of PVB vs. Cntrl samples from the upper as well as the lower lobes (n = 6 for PVB; n = 7 for Cntrl). Using LC-MS/MS-based proteomics, we identified 5112 proteins with high confidence (false discovery rate < 0.01) (Appendix A). The dataset has been submitted to the ProteomeXchange Consortium via the PRIDE partner repository with the dataset identifier PXD038982 [9]. Differential abundance analysis revealed significant differences between groups. The principal component analysis showed clustering of the PVB animals compared to the Cntrls in the upper (HP/HF) lobe, whereas similar clustering was absent in the lower lobe (HP/LF) (Figure 2A,B).

In total, 104 proteins were found to be differentially abundant (Benjamini–Hochberg corrected *p*-value < 0.05 and fold change ≥ 1.5) between PVB and Cntrl in the upper lobes (Appendix A). In the lower lobes, 52 proteins differed significantly in abundance between PVB and Cntrl (Appendix A). Volcano plots were used to visualize proteome alterations between conditions (Figure 3A,B). In the upper lobe, several apolipoproteins (APOF, APOA, APOC3), complement cascades (C6, C7, C8, C9) and coagulation proteins (Serpins, HRG, PROC) were found to be downregulated in the PVB animals, while several membrane transport proteins (SCGB1A1, SFTPA1) were found to be upregulated. In the lower lobe, DNA binding and ribosomal proteins (H1-2, H1-1, RPS6) were upregulated, while membrane remodeling proteins (GMPR, ALF1, SIGLEC1) were downregulated.

We performed a STRING preranked functional enrichment analysis of proteome profiles from the upper and lower lobe to reveal lobe-specific signatures for PVB and Cntrl animals. From the Gene Ontology (GO) biological processes database, 67 and 7 significantly enriched terms were found in the upper and lower lobe, respectively (enrichment factor >1) (Figure 3C, Appendix A), PVB upper lobes showed a distinct downregulation of proteins related to humoral immune regulation, lipoprotein particle organization, cholesterol esterification and triglyceride homeostasis and an upregulation of proteins related to platelet degranulation, coagulation, cholesterol efflux and intermembrane lipid transport. Proteins related to the extracellular matrix were found to be both up- as well as downregulated in PVB animals, indicating altered matrix turnover. In the lower lobe, PVB showed fewer enriched pathways compared to the upper lobe. The majority of the pathways were related to blood coagulation, extracellular matrix reorganization, actin cytoskeletal organization and carbohydrate metabolism. Since ECM-related proteins were altered in both lobes, we also compared the proteins in this pathway in both upper and lower PVB lobes with the established lung matrix gene set and found several proteins (collagens, serpins, etc.) that were differentially regulated (Appendix A) [10].

### 2.3. Metabolomic Profiling of the Lower and Upper Lobe versus Control

Next, we performed untargeted metabolomics on these lung tissues to better understand the ongoing metabolic alterations caused by variations in flow and pressure. To detect the relevant metabolites, we used statistical analysis with XCMS and MetaboAnalyst 5:0 software. Appendix A lists the metabolites that were detected in the HILIC-negative mode MS. On the MetaboAnalyst platform, a 3D PCA analysis (Figure 4A,B) and a supervised orthogonal partial least squares discriminant analysis (OPLS-DA) (Figure 4C,D) were performed for both the upper and lower lobes. In the upper lobe, 3D PCA and OPLS-DA analysis revealed separation between the PVB and Cntrl groups. Similar to the case for our proteomics findings, the lower lobe groups did not show a clear separation in metabolomics either. The OPLS-DA analysis also allowed for the identification of the metabolites that contributed the most to group segregation, known as variable importance in the projection (VIP) scores, and they were ranked accordingly (Figure 5A). Metabolites with a VIP score of ≥1 were interpreted as highly influential (Appendix A), and we performed an enrichment analysis of metabolites with *p* < 0.05 to differentiate control from PVB animals (Figure 5B). In comparison to Cntrl, we found 82 such metabolites in the PVB upper lobe (Appendix A) and 29 metabolites in the PVB lower lobe (Appendix A). Enrichment analysis for the PVB upper lobe revealed 25 metabolic pathways, of which the following six pathways had a *p*-value of <0.05: linoleic acid metabolism, ubiquinone biosynthesis pathway, transfer of acetyl groups into mitochondria, arginine, proline metabolism and glycerolipid metabolism. The lower lobe showed enrichment of 11 pathways, but none were significantly altered (*p* < 0.05). We detected the pathway for purine metabolism (*p* = 0.06) to be the most differentially regulated (Appendix A).

### 2.4. Network Analysis of Proteomics and Metabolomics Datasets

A combined analysis of the two omics datasets was carried out in order to identify commonly altered pathways and to provide additional insight into the process of pulmonary vascular remodeling. The metabolite–metabolite and the gene–metabolite interaction networks provide an overview of functionally related metabolites and proteins found to be most differentially abundant in metabolomics and proteomics. The metabolite–metabolite pathway interaction network derived from the KEGG database is shown in Figure 6A and highlights functional interactions among the top altered metabolites such as oleic acid, linoleic acid, palmitic acid, prostaglandin E2 and L-malic acid, butyric acid, NADP, proline, threonine, S-adenosylhomocysteine and arachidonic acid. Next, the most significantly altered proteins and metabolites identified were mapped to the gene–metabolite molecular interactions to create a network (Figure 6B). The network includes 31 nodes (protein, metabolites) and shows that the metabolites (squares) are upregulated whereas the proteins (filled circles) are downregulated. The metabolite chondroitin sulfate, a major component of the extracellular matrix (ECM), is upregulated in the upper lobe of the PVB group and formed a network with proteins important for wound healing such as serpinc1, serpinD1, F12, PROC, VTN, AMBP and TNC. Plasminogen (PLG), another prominent protein involved in wound healing and ECM remodeling, is functionally linked to both chondroitin sulfate and oleic acid. The PVB upper lobe was enriched in oleic acid, linoleic acid, palmitic acid, butyric acid and arachidonic acid, which formed a network with downregulated proteins in the PVB group such as ApoA1, ApoB, AdipoQ and ALB. These proteins and metabolites together participate in fatty acid metabolism. Prostaglandin E2, a common byproduct of arachidonic acid, is also upregulated in the PVB upper lobe and has formed a network with complement cascade members C8A and C3 as well as the chemokine PPBP, which are involved in inflammation. Finally, the monosaccharide metabolite glucose was downregulated in the upper lobe and functionally linked with the surfactant protein SFPTD and the blood coagulation protein HBB, indicating altered glucose metabolism. The PVB lower lobe presented with a metabolite–metabolite interaction network involving only four metabolites: guanosine monophosphate, inosinic acid, glyceric acid and dodecanoic acid (Appendix A). The gene metabolite network showed a simple network involving only guanosine monophosphate and inosinic acid (Appendix A). They functionally connected with the downregulated the enzyme guanosine monophosphate reductase (GMPR) and HPRT1 in the PVB lower lobe. Furthermore, the metabolite guanosine monophosphate was connected to the interferon-induced guanylate binding protein (GBP1).

### 2.5. Protein and Transcriptional Regulation of Fatty Acid Uptake in the Upper Lobe

The protein abundance of members of apolipoproteins in the upper lobe as well as the lower lobe was further compared in dot plot analysis, and members of this fatty acid uptake pathway were validated at a transcriptional level (Figure 7A–C, Appendix A). ApoE was found to be significantly altered in both proteomics as well as at the transcriptional level in the upper lobe of the PVB group. In addition, further transcripts coding for proteins important for fatty acid uptake such as CD36 (*p* = 0.1) showed a trend toward a decrease in the PVB upper lobe compared to Cntrls along with significant upregulation of the LDLR (low-density lipoprotein receptor). In contrast, we did not see similar changes at the mRNA level for these proteins in the lower lobe of PVB compared to Cntrls.

## 3. Discussion

The main findings in this study were that (i) chronic alterations in flow and pressure induced by PVB impact the proteomic and metabolomic profile in the lung tissue and result in increased ECM and collagen production in lobes with both HP/HF and HP/LF; (ii) upper lung lobes with HP/HF adapt by altering the fatty acid metabolism as well as ROS signaling and (iii) the lower lobes with HP/LF increase their purine metabolism in order to cope with the increased demand of cellular proliferation (Figure 8).

We have previously demonstrated that PH caused by pulmonary vein stenosis results in a progressive increase in pulmonary vascular resistance, which is accompanied by functional (increased contribution of endothelin, phosphodiesterase 5) as well as structural (increased media thickness) pulmonary vascular remodeling [7,11]. Banding of the confluence of veins from the lower lobes results in areas of the lungs with distinct hemodynamic profiles: HP/HF in the unbanded upper lobes and HP/LF in the banded lobes. This model is, therefore, well suited for the study of mechano-metabolic coupling and its role in pulmonary vascular remodeling in PH, as it has been demonstrated that metabolic and structural changes are coupled to each other [12]. Here, we present a thorough proteome and metabolome profile analysis of lung tissue with these distinct mechanical profiles. Pathway enrichment analysis in PVB animals demonstrated changes in several pathways that have been associated with the progression of PH. Thus, alterations were observed for extracellular matrix proteins involving integrins, matrix metalloproteases, collagen, vitronectin, serpins and others observed in both lobes of the animals. This is in accordance with our histological data, suggesting increased ECM deposition around the vessels. We also detected high amounts of phosphatidylcholine (PC) as well as prostaglandins, which indicate plasma membrane break and inflammatory signaling due to high shear stress [13]. In line with these findings, the comparison with the lung matrix database revealed that further extracellular matrix proteins were altered in the upper and lower lobes of PVB swine (Appendix A). ECM proteins such as FGB, COL1A and COL15A1 were significantly upregulated in the PVB lower lobe, whereas in the upper lobe, the analysis revealed synergistic downregulation of extracellular proteolytic proteins such as MMP9 and serpins [14,15].

Strikingly, proteins of the apolipoprotein family were significantly altered in abundance in PVB animals. The key protein component of HDL-C, apolipoprotein A (APOA), which was downregulated in the upper lobe, was not shown to be differentially regulated in the lower lobe. Downregulation of APOA1 is in accordance with data showing that ApoA-1 is less prevalent in PH, which contributes to oxidative stress and endothelial dysfunction [16]. Furthermore, administration of a peptide mimetic of ApoA-1 reduced pulmonary hypertension in rodent models with PH [17]. Along with ApoA, we also detected significantly reduced levels of ApoE at the proteomics as well as at the transcription level in the PVB upper lobe. The metabolomics data in the upper lobe further point to an ongoing alteration of lipid homeostasis and detected increased fatty acids such as oleic acid, linoleic acid, arachidonic acid and palmitic acid in the PVB group, indicating a reduced uptake of fatty acids due to decreased levels of apolipoproteins [18]. High amounts of linoleic and oleic acids have been found to significantly lower nitric oxide (NO) levels in endothelial cells and exert their deleterious effects via ROS [11,19,20]. It has been shown that HIF-1α activation, a common dysregulated pathway in PH and lung diseases, can inhibit β-oxidation of long-chain fatty acids leading to accumulation of fatty acids [21]. However, we did not detect increased accumulation of carnitine and acyl-carnitine, which reflects inhibition of mitochondrial fatty acid β-oxidation and has been previously shown to be involved in the development of PH [22].

In keeping with studies from Umar et al. who showed higher oxidized LDL in the lungs and plasma in PH with a decrease in CD36, our proteomic analysis did detect modifications of pathways regulating cholesterol levels consisting primarily of the downregulation of fatty acid transporters such CD36 and LDLRAP1 [23]. Along with this, we detected significant upregulation of the protein LDLR in the PVB upper lobe. LDLR mainly binds to apolipoprotein B100 (APOB) and APOE to clear cholesterol from the blood [24]. Both ApoB and ApoE are high-affinity ligands for LDLR and are expressed in various immune and vascular cells [24,25]. Negative feedback inhibition from transcriptional and posttranscriptional mechanisms closely controls the LDLR pathway, and disruption of this tightly controlled pathway can influence lipid and cholesterol regulation [26]. These data are also in accordance with integrated proteomic and metabolomics data on HUVECs presented by Venturini et al., showing that high shear stress upregulates the lipoprotein metabolism and increases the expression of LDLR [27].

Additionally, we found metabolites such as oxaloacetic acid and L-malic acid, both intermediate products of the TCA cycle, to be enriched in the PVB upper lobe along with decreased glucose. These metabolites take part in anaplerotic reactions in which the intermediate metabolites exit the TCA cycle and are used by proliferating cells due to an increased demand for protein and fatty acids in PH [12]. These data support the presence of the Warburg effect, showing that glucose metabolism is increased in PH [28,29,30,31]. Further evidence for this Warburg effect is the lower amount of NADP in the PVB upper lobe. NADP maintains the redox balance in the cells and supports the biosynthesis of the fatty acids and is essential for maintaining a large number of biological processe [32]. In agreement with this finding, Nukula et al. reported a lower NADPH/NADP ratio in CTEPH patients’ endothelial cells compared to healthy subjects, implying increased oxidative stress and endothelial cell dysfunction [30].

A key metabolite that was downregulated in the PVB upper lobe and deemed important from our network analysis was S-adenosylhomocysteine (SAH). Asymmetric dimethyl arginine (ADMA), a negative regulator of endothelial nitric oxide synthase, is formed by the hydrolysis of methylated proteins, and the methylated proteins are derived when S-adenosyl methionine (SAM) is converted to SAH. We also simultaneously observed increased aspartic arginine (VIP > 1, Appendix A), which is a source of NO in endothelial cells, in the upper lobe of the PVB group [23,24]. In our previous work, we have shown that NO production is increased in HP/HF areas, likely as a compensatory mechanism to maintain vasodilation [33]. Our current data suggest that low SAH, and hence low ADMA, in combination with high arginine, the substrate for endothelial NO synthesis, facilitates NO synthesis in the PVB upper lobe vasculature.

Notably, proteomic and metabolic alterations were less pronounced between PVB and Cntrls in the lower lobes. The STRING analysis points to reduced glucose synthesis in the PVB lower lobes, which supports the notion that glycolysis predominates over other metabolic activities in PH in the lower lobes as well [22]. Another intriguing observation was that the purine pathway metabolites adenosine monophosphate (AMP) and guanosine monophosphate (GMP) were significantly enriched, and the enzyme guanosine monophosphate reductase (GMPR), which converts GMP to inosine monophosphate (IMP), was downregulated. Additionally, our proteome data revealed that the PVBs had a decreased abundance of the enzyme HPRT, which transforms hypoxanthine into IMP and is crucial for the salvage pathway for recycling nucleotides [34]. We also found more inosine in the metabolome of PVB lower lobes, which indicated that the cells increased de novo purine production rather than using the standard active salvage pathway. These data are consistent with the study by Hautbergue et al., wherein modifications to the purine metabolic pathway in the right ventricle and plasma of PH rats were shown [35]. The purine metabolite levels in endothelial cells from PAH patients have also been found to be higher, the same was true for the serine to glycine ratio, which is mediated by the mitochondrial enzyme serine hydroxymethytransferase (SHMT) [36]. Although SHMT was unchanged in our lung tissue samples, we did observe an increase in the metabolite serine (VIP > 1) in the PVB lower lobes (Appendix A). Moreover, in atherosclerosis models, it has been shown that vessels with low flow and shear stress have decreased endothelial nitric oxide synthase (eNOS) along with increased cell proliferation and collagen deposition [31].

## 4. Materials and Methods

Lung tissue was used from experiments that have previously been published [7,11,33]. These experiments followed the guiding principles in the care and use of laboratory animals, which are endorsed by the Council of the American Physiological Society, and the protocol was approved by the Animal Care Committee at Erasmus University Medical Center (EMC3158, 109-13-09).

### 4.1. Outline of Study

For all surgical procedures, swine were sedated with an intramuscular injection of a mixture of tiletamine/zolazepam (5 mg kg^−1^, Virbac, Barneveld, The Netherlands), xylazine (2.25 mg kg^−1^, AST Pharma, Oudewater, The Netherlands) and atropine (0.5 mg) and intubated and ventilated (O_2_:N_2_ (1:2)). Isoflurane (2% vol/vol, Pharmachemie, Haarlem, The Netherlands) was added to the gas mixture to induce anesthesia. Post-surgical analgesia was administered by means of an i.m. injection (0.3 mg buprenorphine i.m. Indivior, Slough, UK) and a fentanyl slow-release patch (6 or 12 μg h^−1^ depending on body weight, 72 h).

Crossbred Landrace x Yorkshire pigs of either sex (8 ± 2 kg) underwent non-restrictive inferior pulmonary vein banding (n = 6) via the third right intercostal space or a sham procedure (n = 7). All 13 animals, underwent chronic instrumentation 4 weeks later, enabling hemodynamic assessments on awake animals. Following a left-sided thoracotomy in the fourth intercostal space, fluid-filled catheters (Braun Medical Inc., Bethlehem, PA, USA), were inserted in the aorta, the pulmonary artery, the left and right ventricle and the left atrium for the measurement of blood pressure. A flow probe (20PAU, Transonic systems, Ithaca, NY, USA) was placed around the ascending aorta for the measurement of cardiac output. Aorta flow was indexed to bodyweight. The total pulmonary vascular resistance index was calculated as the ratio of mean PAP and cardiac index, while pulmonary vascular compliance was calculated as stroke volume index/(systolic PAP − diastolic PAP). Hemodynamics were recorded (WinDaq, Dataq Instruments, Akron, OH, USA) in the awake state, with swine standing quietly, and analyzed offline using a custom written program (Matlab, version R2007b, The MathWorks).

Twelve weeks after the PVB procedure, swine were re-anesthetized; the thorax was opened using sternotomy, and the heart and lungs were excised, snap-frozen in liquid nitrogen and processed for further analysis.

### 4.2. Real-Time Quantitative PCR of Lung Tissue

Lung tissue was snap-frozen and 30 mg of tissue was homogenized, and mRNA was extracted using the RNeasy Fibrous Tissue Mini kit (Qiagen, Hilden, Germany). cDNA was synthesized using 500 ng of mRNA and the SenSi FAST cDNA synthesis kit (Bioline, London, UK). Target genes were normalized against beta-actin and cyclophilin using the CFX manager software 3.1 (BioRad, CA, USA). Relative gene expression was calculated using the delta–delta Ct method.

### 4.3. Proteomics

#### 4.3.1. Sample Preparation for Proteome Analysis

Frozen lung tissue samples were placed into precooled tubes and cryopulverized in a CP02 Automated Dry Pulverizer (Covaris, Woburn, MA, USA) with an impact level of 5 according to the manufacturer’s instructions. Tissue lysis was performed in 8 M urea/0.5 M NH_4_HCO_3_ with ultrasonication (18 cycles of 10 s) using a Sonopuls HD3200 (Bandelin, Berlin, Germany). Total protein concentration was quantified using a Pierce 660 nm Protein Assay (Thermo Fisher Scientific, Rockford, IL, USA). Fifty micrograms of protein were digested sequentially, firstly with Lys-C (FUJIFILM Wako Chemicals Europe GmbH, Neuss, Germany) for 4 h and, subsequently, with modified porcine trypsin (Promega, Madison, WI, USA) for 16 h at 37 °C.

#### 4.3.2. Nano-Liquid Chromatography (LC)–Tandem Mass Spectrometry (MS) Analysis and Bioinformatics

1 μg of the digest was injected on an UltiMate 3000 nano-LC system coupled online to a Q Exactive HF-X instrument operated in the data-dependent acquisition (DDA) mode. Peptides were transferred to a PepMap 100 C18 trap column (100 µm × 2 cm, 5 µM particles, Thermo Fisher Scientific) and separated on an analytical column (PepMap RSLC C18, 75 µm × 50 cm, 2 µm particles, Thermo Fisher Scientific) at a 250 nL/min flow rate with a 160 min gradient of 3–25% of solvent B followed by a 10 min ramp to 40% and a 5 min ramp to 85%. Solvent A consisted of 0.1% formic acid in water and solvent B of 0.1% FA in acetonitrile. MS spectra were acquired using a top-15 data-dependent acquisition method on a Q Exactive HF-X mass spectrometer. Protein identification was carried out using MaxQuant (v.1.6.7.0) [37] and the NCBI RefSeq Sus scrofa database (v.7-5-2020). All statistical analyses and data visualization were performed using R (https://www.r-project.org/) (accessed on 29 December 2022). Prior to statistical analysis, potential contaminants, only identified by site and reverse hits were excluded. Proteins with at least two peptides detected in at least three samples of each condition were quantified using the MS-EmpiRe algorithm as previously described [38,39]. The R script used for quantitative analysis is available at https://github.com/bshashikadze/pepquantify (accessed on 7 September 2022). Proteins with a Benjamini–Hochberg corrected *p*-value ≤ 0.05 and fold change ≥ 1.5 were regarded as significantly altered. Preranked gene set enrichment analysis using STRING was employed to reveal biological processes related to differentially abundant proteins [40]. Signed (based on fold change) and log-transformed *p*-values were used as ranking metrics and the false discovery rate was set to 1%. The redundancy of the significantly enriched biological processes was minimized using REVIGO tool [41].

### 4.4. Metabolomics

Approximately 50 mg of sample material was weighed in a 2 mL bead beater tube (CKMix, Bertin Technologies, Montigny-le-Bretonneux, France) filled with 2.8 mm and 5.0 mm ceramic beads. Then, 1 mL of a methanol/water mixture (70/30, *v*/*v*) was added, and the samples were extracted with a bead beater (Precellys Evolution, Bertin Technolgies, Montigny-le-Bretonneux, France) supplied with a Cryolys cooling module 3 times each for 20 s with 15 s breaks in between at 10,000 rpm. After centrifugation at 13,000 U/min for 10 min, the supernatants were dried by vacuum centrifugation, suspended in 150 µL of methanol/water (70/30, *v*/*v*) and subjected to MS analysis.

Untargeted analysis was carried out on a Nexera UHPLC system connected to a Q-TOF mass spectrometer (TripleTOF 6600, AB Sciex, MA, USA). Chromatographic separation was achieved by using a HILIC UPLC BEH Amide 2.1 × 100, 1.7 µm column with a 0.4 mL/min flow rate. The mobile phase consisted of 5 mM ammonium acetate in water (eluent A) and 5 mM ammonium acetate in acetonitrile/water (95/5, *v*/*v*) (eluent B). The following gradient profile was used: 100% B from 0 to 1.5 min, 60% B at 8 min, 20% B at 10 min to 11.5 min and 100% B at 12 to 15 min. Aliquots of 5 µL per sample were injected into the UHPLC-TOF-MS. The autosampler was cooled to 10 °C, and the column oven was heated to 40 °C. A quality control (QC) sample was pooled from all samples and injected after every 10 samples. MS settings in the positive mode were as follows: gas 1 55, gas 2 65, curtain gas 35, temperature 500 °C, ion spray voltage 5500, declustering potential 80. The mass range of the TOF-MS scans was 50–2000 *m*/*z*, and the collision energy was ramped from 15 to 55 V. MS settings in the negative mode were as follows: gas 1 55, gas 2 65, cur 35, temperature 500 °C, ion spray voltage −4500, declustering potential −80. The mass range of the TOF-MS scans was 50–2000 *m*/*z*, and the collision energy was ramped from −15 to −55 V.

The “msconvert” tool from ProteoWizard [42] was used to convert raw files to mzXML (denoised by centroid peaks). The bioconductor/R package xcms [43] was used for data processing and feature identification. More specifically, the matched filter algorithm was used to identify peaks (full width at half maximum set to 7.5 s). Then the peaks were grouped into features using the “peak density” method. The area under the peak was integrated to represent the abundance of features. The retention time was adjusted based on the peak groups presented in most samples. To annotate features with the names of metabolites, the exact mass and MS2 fragmentation pattern of the measured features were compared to the records in HMBD [44] and the public MS/MS spectra in MSDIAL [45], referred to as MS1 and MS2 annotation, respectively. Missing values were imputed with half of the limit of detection (LOD) methods, i.e., for every feature, the missing values were replaced with half of the minimal measured value of that feature in all measurements. To confirm that an MS2 spectrum was well annotated, we manually reviewed our MS2 fragmentation pattern and compared it with records in the public database or previously measured reference standards to evaluate the correctness of the annotation.

The MetaboAnalyst 5.0 platform was utilized to conduct multivariate data analysis, for PCA and OPLS-DA. The contribution of each variable to the classification was indicated by the VIP value that was calculated in the OPLS-DA model after Pareto scaling. The Student’s *t*-test at the univariate level was further employed to measure the significance of metabolites with VIP > 1.0. Metabolites with a *p*-value < 0.1 were considered as differential metabolites, while those with a *p*-value < 0.05 were recognized as statistically significant differential metabolites. Enrichment analysis and network analysis was performed using only the significant metabolites and significant genes using the KEGG pathway database.

## 5. Conclusions

In conclusion, our combined omics study showed PVB-related key metabolic alterations in a compartment-specific manner. The combination of a model of PH, with specific changes in shear stress in different areas of the lung, with proteome and metabolomic data shows that particular metabolic pathways, including fatty acid absorption and purine synthesis, are altered in early PH. Such a deeper understanding of the metabolic changes in lung tissue may provide new targets for therapy and may, thereby, pave the way for new avenues in precision medicine for PH.

## Figures and Tables

**Figure 1 ijms-24-04870-f001:**
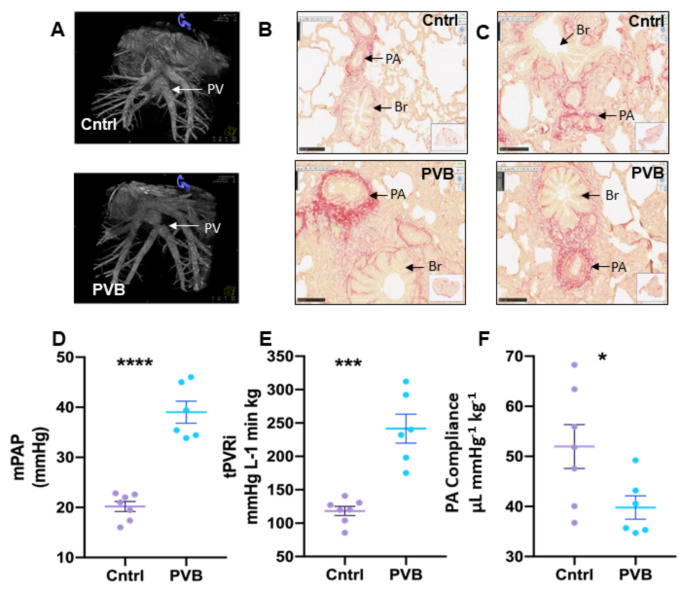
Pulmonary vein banding causes PH. (**A**) Angiogram of the inferior venous confluence of a representative Cntrl (top) and PVB animal (bottom). (**B**,**C**) Picrosirius red staining of the lung upper and lower lobe in Cntrl and PVB animals. (**D**–**F**) Mean pulmonary artery pressure (mPAP), total pulmonary vascular resistance index (tPVRi) and pulmonary artery compliance in awake, resting swine. PA—pulmonary artery, PV—pulmonary vein, Br—bronchiole. Values are means ± SEM. * *p* ≤ 0.05, *** ≤0.001, **** ≤0.0001. Scale bar: 100 μm. PVB vs. Cntrl by Student’s *t*-test, Cntrl n = 7; PVB n = 6.

**Figure 2 ijms-24-04870-f002:**
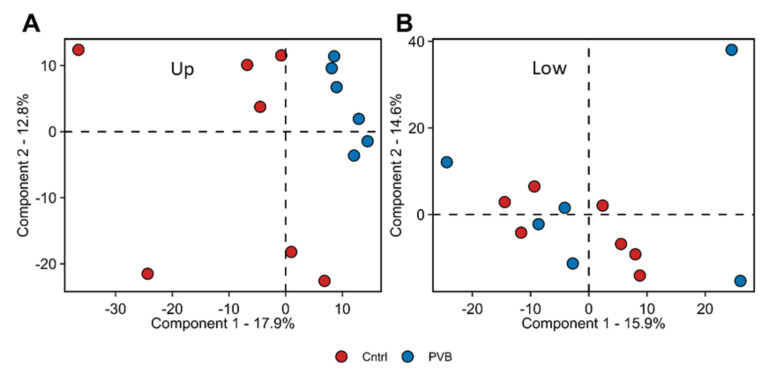
Proteomic analysis of lung tissue of PVB vs. Cntrl animals. (**A**,**B**) Principal component analysis (PCA) of proteome profiles from upper (**A**) and lower (**B**) lobe of Cntrl (n = 7) and PVB animals (n = 6). Colored circles indicate individual animals. Red-filled circles are Cntrl and blue-filled circles are PVB animals.

**Figure 3 ijms-24-04870-f003:**
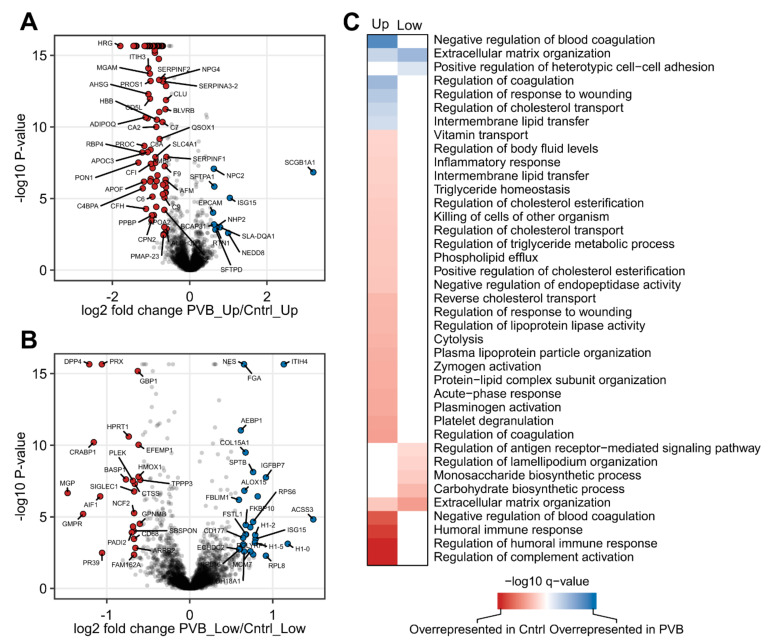
Quantitative and functional analysis of upper and lower lobe of PVB vs. Cntrl animals. (**A**,**B**) The quantitative proteome alterations are visualized via volcano plots in upper and lower lobes, respectively, from PVB vs. Cntrl pigs. Color-filled circles (blue—upregulated, red—downregulated) indicate differentially abundant proteins (Benjamini–Hochberg corrected *p*-value ≤ 0.05 and fold change ≥ 1.5). (**C**) Functional characterization of differences between upper and lower lobes in PVB and Cntrl animals. Heatmap shows GO-term enrichment in PVB compared to Cntrl. Cntrl (n = 7); PVB (n = 6).

**Figure 4 ijms-24-04870-f004:**
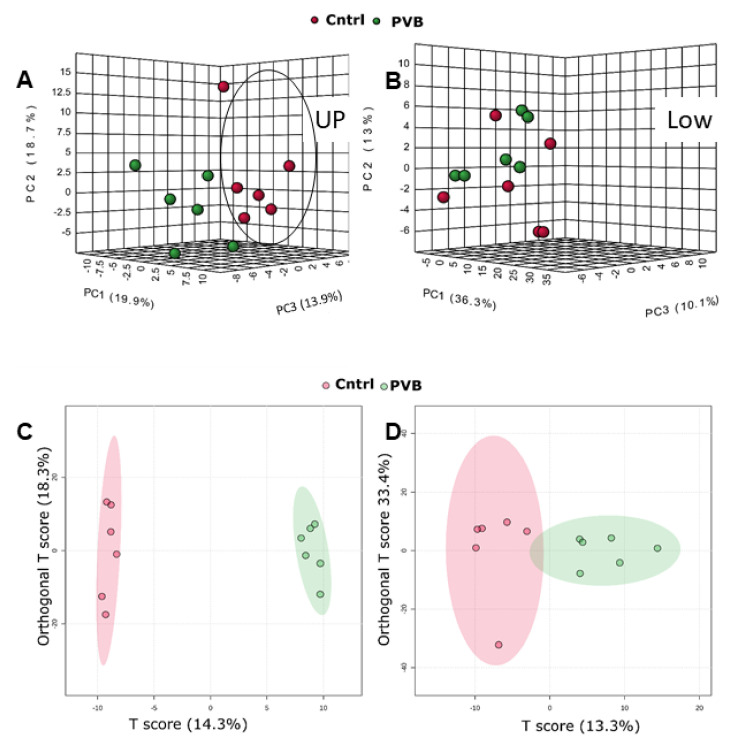
Metabolomic analysis in the upper and lower lobe of PVB vs. Cntrl animals. (**A**,**B**) Three-dimensional PCA analysis based on metabolic profiles of PVB and Cntrl animals in the upper and lower lobe in negative mode (HILIC). (**C**,**D**) OPLS-DA analysis based on metabolic profiles of PVB and Cntrl animals in upper and lower lobe in negative mode (HILIC). n = 6 in each group.

**Figure 5 ijms-24-04870-f005:**
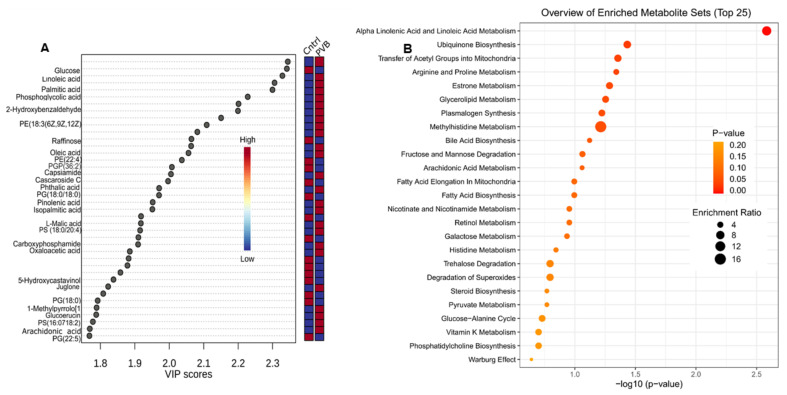
Metabolomic analysis in the upper lobe of PVB vs. Cntrl animals. (**A**) Metabolites ranked by Variable importance in the projection score (VIP) > 1 in PVB vs. Cntrl pigs in upper lobes based on OPLS-DA analysis. (**B**) Pathway analysis based on KEGG database of the enriched metabolites (VIP > 1 and *p*-value < 0.05 in the upper lobe of PVB vs. Cntrl swine. n = 6 in each group. The size of the dots represents the enrichment ratio and the shade of the color represent the −log 10 (*p* value).

**Figure 6 ijms-24-04870-f006:**
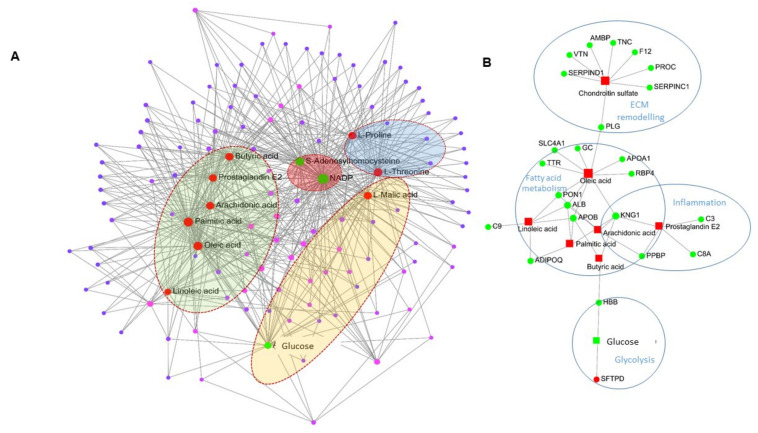
Interaction network analysis of the metabolites and proteins from metabolomics and proteomics. (**A**) Metabolite–metabolite interaction network for the most enriched metabolites in the upper lobe of PVB vs. Cntrl. Labeled and red-filled circles represent the upregulated metabolites, and green-filled circles represent the downregulated metabolites detected in the PVB group. Metabolites clustered in green background represent metabolites participating in fatty acid metabolism, red background—ROS pathway; blue—ECM remodeling and yellow—glucose metabolism. (**B**) Gene–metabolite network analysis for the significantly altered proteins and enriched metabolites for upper lobe of PVB vs. Cntrl. The red-filled squares represent the downregulated proteins, and green-filled circles represent the upregulated metabolites detected in the PVB group. Metabolites of VIP > 1 with *p*-value < 0.05 and proteins with fold change > 1.3 and Benjamini–Hochberg-adjusted *p*-value < 0.05 were used for the analysis. The proteins and metabolites belonging to one GO term are encircled together.

**Figure 7 ijms-24-04870-f007:**
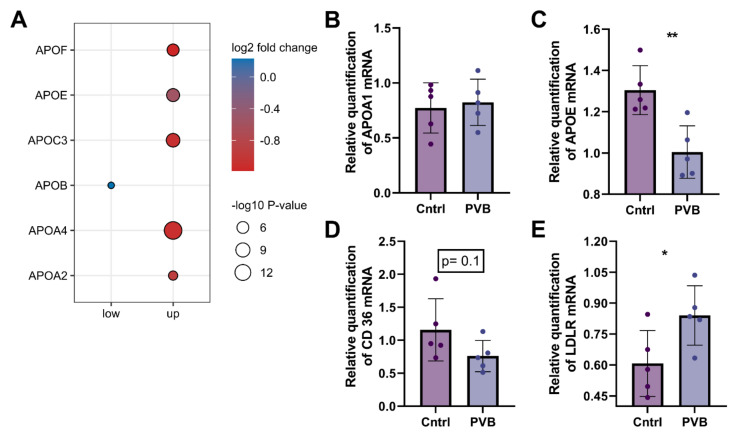
Fatty acid uptake pathway in the upper lobe at the protein and mRNA level. (**A**) Protein abundance of apolipoproteins that were significantly changed (Benjamini–Hochberg corrected *p*-value ≤ 0.05) between PVB compared to Cntrls in the upper and lower lobes are depicted as dots. n = 6 per group. (**B**–**E**) mRNA level of APOA, APOE, CD36 and LDLR, respectively, are expressed relative to the mean expression of the sham animals (values are mean +/− SEM. * *p* < 0.05, ** *p* < 0.01 Students’s *t*-test n = 5 animals per group).

**Figure 8 ijms-24-04870-f008:**
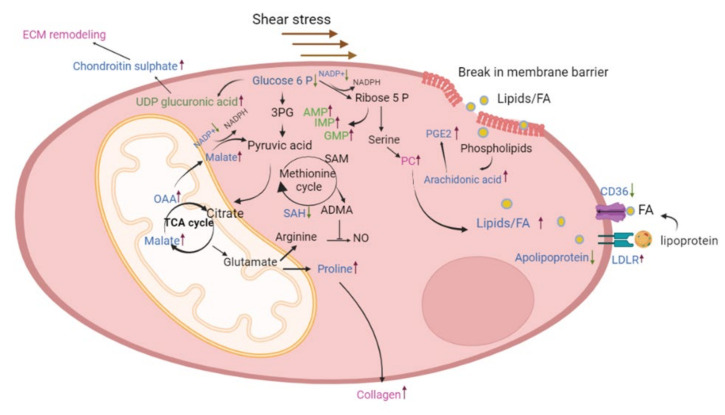
Schematic illustration of the metabolic pathways altered in PVB. The blue-labeled metabolites and proteins are differentially expressed in the upper lobe, which is under high shear stress, and the green-labeled ones are differentially expressed in the lower lobe, which is under low shear stress. Pathways or metabolites/proteins labeled in red are found to be altered in both the lobes. 3PG—3-phosphoglyceric acid, SAM—S-adenosyl methionine, SAH—S-adenosylhomocysteine, NO—nitric oxide, PC—phosphatidyl choline, OAA—oxaloacetic acid.

## Data Availability

The mass spectrometry proteomics dataset has been deposited to the PRIDE repository, dataset identifier PXD038982. The untargeted metabolomics data have been uploaded to the MassIVE database https://massive.ucsd.edu/ProteoSAFe/static/massive.jsp with ID: MSV000090966 (accessed on 24 December 2022).

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
