# Peer review of "Proteomics- and Metabolomics-Based Analysis of Metabolic Changes in a Swine Model of Pulmonary Hypertension"

_ijms, 2023, doi:10.3390/ijms24054870_

Round 1
Reviewer 1 Report
This is a basic study regarding PH due to pulmonary vein banding (PVB), which conducted a quantitative LC-MS/MS based proteomic analysis of lung samples along with untargeted metabolomics from swine with PVB and Control swine, and analyzed the tissues from upper and lower lobes to investigate how different hemodynamic profiles impact protein and metabolite expression due to PH in the lobes. The authors detected changes in the upper lobes for the PVB animals mainly pertaining to fatty acid metabolism, ROS signaling and extracellular matrix remodeling, and small albeit significant changes in the lower lobes for purine metabolism. This reviewer considers that the authors well performed the present study, and has only minor comments as described below.
Minor comments:
1. Figures 1-2 need higher resolution.
2. Histology in Figure 1. The authors should indicate which are pulmonary arteries and which are pulmonary veins.
Author Response
We would like to thank the reviewer for his/ her positive comments. We have increased the resolution of all figures, and added indications of bronchi, and arteries in the histology pictures. It is difficult to distinguish pulmonary small arteries and pulmonary veins in the lung. Typically, the arteries run close to the bronchi and the veins run in the septae, but for the smaller vessels, it is not possible to denote which one is which, as arterioles and venules both have few layers of vascular smooth muscle.
Reviewer 2 Report
In this manuscript, the authors employed unbiased proteomic and metabolomic analyses on both the upper and lower lobes of the swine lungs following pulmonary vein banding in order to identify the regions with metabolic alterations.
The study is well designed and results are straightforward and clearly presented. Overall, the manuscript is well written. I have following comments:
There is a mismatch between the number of animals in Methods (lines 354-356) and figure legends (lines 84-86).
The authors should avoid gratuitous capitalization.
Abbreviations should be defined at first mention in the text only once.
Line 230-233. This text belongs to the reviewer report.
Author Response
We would like to thank the reviewer for his/her positive comments.
There is a mismatch between the number of animals in Methods (lines 354-356) and figure legends (lines 84-86).
We apologize for the oversight and have corrected the number of animals (PVB n=6, Control n =7)
The authors should avoid gratuitous capitalization.
We are not exactly sure what the reviewer means, but have carefully checked and removed unnecessary capitals
Abbreviations should be defined at first mention in the text only once.
We have checked for consistency and only defined abbreviations once.
Line 230-233. This text belongs to the reviewer report.
We have removed these lines